# Molecular Prevalence, Genetic Diversity, and Tissue Tropism of *Bartonella* Species in Small Mammals from Yunnan Province, China

**DOI:** 10.3390/ani14091320

**Published:** 2024-04-28

**Authors:** Pei-Yu Han, Fen-Hui Xu, Jia-Wei Tian, Jun-Ying Zhao, Ze Yang, Wei Kong, Bo Wang, Li-Jun Guo, Yun-Zhi Zhang

**Affiliations:** 1Yunnan Key Laboratory of Screening and Research on Anti-Pathogenic Plant Resources from Western Yunnan, Yunnan Key Laboratory of Zoonotic Disease Cross-Border Prevention and Quarantine, Institute of Preventive Medicine, School of Public Health, Dali University, Dali 671000, China; hanpeiyu1511@gmail.com (P.-Y.H.); xufenhui1569@163.com (F.-H.X.); tjwfatorange@outlook.com (J.-W.T.); zhaojunying0714@163.com (J.-Y.Z.); yangzeyz@163.com (Z.Y.); kongwei4357@163.com (W.K.); 2Department of Biomedical Sciences and Pathobiology, Virginia-Maryland College of Veterinary Medicine, Virginia Polytechnic Institute and State University, Blacksburg, VA 24061, USA; bowang@vt.edu

**Keywords:** *Bartonella*, small mammals, genetic diversity, evolution, tissue tropism

## Abstract

**Simple Summary:**

*Bartonella* is an intracellular parasitic zoonotic pathogen that can infect animals and cause a variety of human diseases. The prevalence and the copy number of *Bartonella* spp. in different tissues in small mammals were studied using conventional PCR and real-time quantitative PCR in Yunnan Province, China. Results showed a 31.5% detection rate, varying across species. Genetic analysis identified thirty, ten, and five strains based on *ssrA*, *rpoB*, and *gltA* genes, with nucleotide identities ranging from 92.1% to 100.0%. Seven species of *Bartonella* were identified, including *B. grahamii*, *B. rochalimae*, *B. sendai*, *B. koshimizu*, *B. phoceensis*, *B. taylorii*, and a new species in *Episoriculus leucops* (GS136). Analysis of the different tissues naturally infected by *Bartonella* spp. revealed varied copy numbers across different tissues, with the highest load in spleen tissue. These findings underscore *Bartonella*’s species diversity and host range in Yunnan Province, highlighting the presence of extensive tissue tropism in *Bartonella* spp. naturally infecting small mammalian tissues.

**Abstract:**

*Bartonella* is an intracellular parasitic zoonotic pathogen that can infect animals and cause a variety of human diseases. This study investigates *Bartonella* prevalence in small mammals in Yunnan Province, China, focusing on tissue tropism. A total of 333 small mammals were sampled from thirteen species, three orders, four families, and four genera in Heqing and Gongshan Counties. Conventional PCR and real-time quantitative PCR (qPCR) were utilized for detection and quantification, followed by bioinformatic analysis of obtained DNA sequences. Results show a 31.5% detection rate, varying across species. Notably, *Apodemus chevrieri, Eothenomys eleusis, Niviventer fulvescens, Rattus tanezumi, Episoriculus leucops, Anourosorex squamipes*, and *Ochotona Thibetana* exhibited infection rates of 44.4%, 27.7%, 100.0%, 6.3%, 60.0%, 23.5%, and 22.2%, respectively. Genetic analysis identified thirty, ten, and five strains based on *ssrA*, *rpoB*, and *gltA* genes, with nucleotide identities ranging from 92.1% to 100.0%. *Bartonella* strains were assigned to *B. grahamii*, *B. rochalimae*, *B. sendai*, *B. koshimizu*, *B. phoceensis*, *B. taylorii*, and a new species identified in *Episoriculus leucops* (GS136). Analysis of the different tissues naturally infected by *Bartonella* species revealed varied copy numbers across different tissues, with the highest load in spleen tissue. These findings underscore *Bartonella*’s diverse species and host range in Yunnan Province, highlighting the presence of extensive tissue tropism in *Bartonella* species naturally infecting small mammalian tissues.

## 1. Introduction

*Bartonella* constitutes a group of aerobic and microaerobic Gram-negative bacteria that parasitise animal cells. Taxonomically, it falls under subgroup α2 of the class Alphaproteobacteria, order Hyphomicrobiales, family *Bartonellaceae*, genus *Bartonella*. There are a total of 39 species and three subspecies with valid taxonomic status in this genus (https://www.bacterio.net/genus/bartonella, assessed on 14 January 2024). Seventeen of these *Bartonella* species or subspecies have been confirmed to be associated with human diseases [1], with six transmitted by rodents recognized as human pathogens, including *Bartonella grahamii* [2], *Bartonella elizabethae* [3], *Bartonella tribocorum* [4], *Bartonella vinsonii subsp. Arupensis* [5], *Bartonella washoensis* [6], and *Bartonella tamiae* [7]. These pathogens collectively cause bartonellosis, leading to various human illnesses such as cat-scratch disease, Carrion’s disease, chronic lymphadenopathy, endocarditis, trench fever, chronic bacteraemia, bacillary angiomatosis, bacillary peliosis, vasculitis, uveitis, and others [1]. Globally, as of 2022, 24 cases of rodent-associated *Bartonella* species in humans have been reported [8].

*Bartonella*, as a zoonotic pathogen, is prevalent in both wild and domestic mammals, displaying a considerable degree of prevalence and genetic diversity [9], a trait evolved to enhance evasion of host animal immunity [10]. The first documented case of *Bartonella* spp. infecting a person was “trench fever” during World War I, attributed to the louse-borne *Bartonella quintana* [11]. Currently, *Bartonella* is distributed worldwide, reported in all regions except the Middle East, Central Africa, and Latin America, where countries with low health coverage remain uninvestigated. *Bartonella* infects a diverse array of host animals, primarily including mammals such as Rodentia [12], Lagomorpha [13], Carnivora [14], Chiroptera [15], and Artiodactyla [16,17], as well as Primates (e.g., macaques) [18], Insectivora (e.g., shrews) [19], and certain birds and fish.

Since the 1980s, molecular technology has continually advanced the classification of *Bartonella*, progressing from DNA–DNA hybridization technology to multi-locus sequence analysis (MLSA) and the use of single-gene detection technology. Detected genes encompass 16S rDNA, the 16S-23S rRNA intergenic spacer region (ITS), citrate synthase (*gltA*), genes encoding RNA polymerase subunits (*rpoB*), the riboflavin synthase gene (*ribC*), *groEL*, the cell division protein gene (ftsZ), etc. Notably, the *rpoB* gene has emerged as a reliable, reproducible, and accurate tool for bacterial detection and identification [20]. The *rpoB* gene, being a widespread single-copy gene in bacteria, is characterized by its susceptibility to synonymous substitutions and substantial conservation [21,22]. La Scola et al. conducted a comparative analysis of seven different loci in *Bartonella*, revealing that the *rpoB* and *gltA* genes, utilized for intraspecific differentiation, proved to be the most effective [10]. However, it is noteworthy that homologous recombination events have been observed within the *gltA* gene. In conventional PCR, both the *rpoB* and *gltA* genes are commonly employed for the detection of *Bartonella* spp. Despite the effectiveness of this approach, traditional PCR has its limitations. To overcome the drawbacks associated with traditional PCR, the detection of single-copy prokaryotic specific molecules, such as the *ssrA* gene, has been facilitated through quantitative PCR (qPCR) [23]. This method offers heightened specificity and sensitivity in the *Bartonella* assay, enabling a more precise and accurate assessment of the presence of *Bartonella* species.

In this study, our focus was on assessing the prevalence of *Bartonella* species among small mammals in Yunnan, China. Recognizing the limitations associated with detecting *Bartonella* species primarily through a single gene and employing only one detection method, we implemented a comprehensive approach. Initially, real-time PCR was employed to detect samples, offering a rapid and efficient means of initial screening. Subsequently, conventional PCR was utilized to target different genes, allowing for a more nuanced analysis of *Bartonella* diversity. To augment our investigation, we conducted quantitative analysis on positive samples, assessing their prevalence in various tissues including the heart, lung, liver, spleen, kidney, intestine, and brain. This multifaceted approach aimed to provide a more comprehensive understanding of *Bartonella* species prevalence among small mammals in the specified region.

## 2. Materials and Methods

### 2.1. Sample Collection and Processing

Small mammal samples were collected using live rat traps (20 × 12 × 10.5 cm^3^, Xiangyun Hong Jin Mouse Cage factory, Dali, China) baited with freshly deep-fried dough sticks in Heqing County and Gongshan County in Yunnan Province from August 2020 to August 2022. A total of 200 cages were set every five meters during the evening and collected in the morning, with this process repeated for approximately 6 days in each habitat. Small animals were gently anesthetized in an induction chamber filled with cotton infused with isoflurane (1 mL of isoflurane per 500 mL of chamber volume). Isoflurane is an anaesthetic drug prescribed by the Medical Ethics Committee of Dali University and its main advantage is that it allows the animal to be under anaesthesia with less pain. Following anesthesia, animals were compassionately sacrificed on a warming blanket to minimize distress. Dissections were performed to extract heart, liver, spleen, lung, kidney, brain, and intestinal tissues. These tissues were stored in 2 mL cryogenic vials (CORNING, Shanghai, China) at −80 °C for subsequent analysis. Molecular biological identification involved the amplification of the mitochondrial cytochrome b (mt-Cytb) gene of liver tissue DNA using PCR as described previously [24,25,26,27]. A total of 333 small mammals, comprising thirteen species across three orders, four families, and four genera, were captured in Heqing County and Gongshan County in Yunnan Province (Figure 1).

### 2.2. DNA Extraction

Under aseptic conditions, approximately 1 g samples of heart, liver, spleen, lung, kidney, brain, and intestinal tissues were clipped into GeneReady Animal PIII pulverizing tubes (Life Real, Guangzhou, China). Following the addition of 600 μL of sterilized phosphate buffer solution (PBS), samples were ground using a GeneReady Ultimate grinder (Life Real, Guangzhou, China). Subsequently, 300 μL of supernatant from the ground tissue samples was extracted using a DNA extraction kit (TIANGEN, Beijing, China) with a fully automated nucleic acid extraction and purification instrument (BIOER, Hangzhou, China). Extracted samples were stored at −80 °C for subsequent analysis [28,29].

### 2.3. Bartonella spp. Detection

The species-specific primer *ssrA* (*Bartonella*’s non-coding RNA gene) was selected for fluorescence quantitative real-time PCR of nucleic acids extracted from spleen tissue samples (*ssrA-F*: GCTATGGTAATAAATGGACAATGAAATAA; *ssrA-R*: GCTTCTGTTGCCAGGTG; *ssrA-P*: FAM-ACCCCGCTTAAACCTGCGACG-BHQ1) [23,30,31]. The reaction system was as follows: HR qPCR Master Mix (10 μL), *ssrA-*F and *ssrA-*R (0.4 μM each), RNase-Free ddH_2_O (7.8 μL), DNA template (1 μL), and *ssrA-*P (0.4 μL, 10 µM). The amplification reaction took place using the Applied Biosystems 7500 Real-Time PCR system (Thermo Fisher Scientific, Waltham, MA, USA). The amplification conditions comprised 45 cycles with initial predenaturation at 95 °C for 30 s, followed by denaturation at 95 °C for 10 s, annealing at 53 °C for 1 min, and fluorescence signal acquisition times. To prevent contamination and false positives, sample processing, reaction system preparation, and PCR amplification were carried out in separate areas. Controls included a blank control, a negative control (*Orientia tsutsugamushi*, Seoul *Orthohantavirus*), and a positive control (A synthetic plasmid). Plasmid standards were utilized to establish the standard curve, and results were interpreted based on the curve; a C_t_ value < 35 was deemed positive, while a C_t_ value ≥ 35 was considered negative.

The three genes of *gltA*, *rpoB*, and *ssrA* were amplified using conventional PCR (*gltA-*F: GGGGACCAGCTCATGGTGG, *gltA-*R: AATGCAAAAAGAACAGTAAACA; *rpoB-*F: CGCATTGGCTTACTTCGTATG, *rpoB-*R: GTAGACTGATTAAACGCTG; *ssrA-*F: GCTATGGTAATAA ATGGACAATGAAATAA; *ssrA-*R: GCTTCTGTTGCCAGGTG) [29,30,31]. The *gltA* amplification was carried out with the following conditions: pre-denaturation at 95 °C for 3 min, followed by 35 cycles at 94 °C for 15 s (denaturation), 48 °C for 15 s (annealing), and 72 °C for 30 s (extension), with a final extension at 72 °C for 5 min and a 1 min cooling step at 10 °C. The length of the amplification product after PCR was approximately 379 bp (*gltA*). The *rpoB* amplification was conducted under the following conditions: pre-denaturation at 95 °C for 3 min, followed by 35 cycles at 94 °C for 15 s (denaturation), 48 °C for 15 s (annealing), and 72 °C for 30 s (extension), with a final extension at 72 °C for 5 min and a 1 min cooling step at 10 °C. The length of the amplification product after PCR was approximately 866 bp (*rpoB*). The *ssrA* amplification was conducted under the following conditions: pre-denaturation at 95 °C for 3 min, followed by 35 cycles at 94 °C for 15 s (denaturation), 53 °C for 15 s (annealing), and 72 °C for 30 s (extension), with a final extension at 72 °C for 5 min and a 1 min cooling step at 10 °C. The length of the amplification product after PCR was approximately 301 bp (*ssrA*). The amplified products underwent agarose gel electrophoresis for identification. Positive amplicons, aligning with the expected size, were excised and purified from the gel (OMEGA Bio-tek, Norcross, GA, USA), then forwarded to Shenggong Bioengineering Co., Ltd. (Shanghai, China) for bidirectional sequencing. Utilizing BLASTn and BLASTp analyses, we determined nucleotide sequence similarity and translated amino acid similarity against sequences in the NCBI GenBank database (https://blast.ncbi.nlm.nih.gov/Blast.cgi, accessed on 14 January 2024).

### 2.4. Sequence Identification and Phylogenetic Analysis

The DNA sequences were assembled using the DNAstar Lasergene software package version 7.1.0, employing manual editing and precision trimming to generate the definitive *Bartonella* genome sequence. Subsequently, a thorough similarity analysis was conducted using the BLAST. The representative reference sequences for the *gltA*, *rpoB*, and *ssrA* genes were obtained from the NCBI GenBank database. Sequence alignment was performed using ClustalX2, followed by DNA sequence analysis utilizing the maximum likelihood method in MEGAX 11.0 [32]. The Kimura two-parameter method was employed for phylogenetic analysis. Typically, a self-similarity value greater than 70% is considered a reliable indicator of evolutionary branching. The results were visualized in iTOL (https://itol.embl.de/, accessed on 14 January 2024). The present study utilized all sequences obtained from the three genes of *ssrA*, *gltA*, and *rpoB* in GeneBank (accession numbers: OR993996–OR994040).

### 2.5. Construction of Plasmids and Determination of DNA Copy Number

A partial *ssrA* fragment of *Bartonella* was cloned into the pEASY-T1 vector (TransGen Biotech, Beijing, China), and the resulting T-loaded construct was transformed into DH5α *E. coli* cells. Following bacterial liquid amplification, the presence of the inserted target gene was confirmed through sequencing analysis. A small quantity of plasmids carrying the cloned *ssrA* fragment were extracted using the Plasmid Mini Kit I (Omega Bio-tek, Norcross, GA, USA) and stored at −80 °C for future use. Upon melting, the concentration of *Bartonella* plasmids extracted was determined by an ultraviolet spectrophotometer (Life Real, Hangzhou, China). The concentration was then converted into a copy number using the formula:

Copies/µL = Plasmid concentration (ng/µL) × 10^−9^ × 6.02 × 10^23^/(660 × DNA length)

The plasmid concentration was converted into a copy number, which was used for the establishment of a standard curve and quantitative analysis as the positive control.

### 2.6. qPCR Optimization

The positive standards, serving as templates for optimizing conditions, were diluted with RNase-free ddH_2_O in a ten-fold gradient and stored at −20 °C for future use. The optimization process involved fine-tuning the primers, probe concentration, and annealing temperature within the qPCR system through several trials to determine the optimal amplification conditions and reaction system.

For the qPCR experiments, the HiScript^®^ II U+ One Step qRT-PCR Probe Kits were utilized. After multiple rounds of experimentation to optimize the conditions, the final reaction system for qPCR was as follows: AceQ Universal U+ probe: 10 μL; primers (10 μM): 0.4 μL each; taq probe (10 μM): 0.4 μL; RNase-free ddH_2_O: 7.8 μL; DNA template: 1 μL. The total reaction system for the assay was 20 µL, and the amplification reaction was carried out using the Applied Biosystems 7500 Real-Time PCR system (Thermo Fisher Scientific, Waltham, MA, USA). The amplification conditions included 45 cycles at 95 °C with a 30 s pre-denaturation, followed by denaturation at 95 °C for 10 s, annealing temperatures, and fluorescence signal acquisition times at 51 °C for 34 s.

### 2.7. Standard Curves Plotting

To establish the standard curve for the qPCR assay, the positive standard was subjected to a ten-fold gradient dilution, with each concentration repeated three times. The average of the three repetitions was calculated. The logarithmic value of the positive standard’s copy number served as the horizontal coordinate, while the corresponding C_t_ value of the assay served as the vertical coordinate. The standard curve was plotted, and the slopes and correlation coefficients were determined.

### 2.8. qPCR Sensitivity and Specificity Evaluation

The detection limit of qPCR was assessed through serial dilution of *Bartonella* plasmid DNA, initially at a concentration of 356.828 ng/µL, converted to pre-dilution concentrations of 6.09 × 10^10^ copies/µL. This plasmid was successively diluted through ten-fold dilutions with RNase-free H_2_O to serve as the qPCR template. The primer’s specificity was evaluated by referencing the relevant literature. It demonstrated the ability to detect all types of *Bartonella*. This was confirmed by comparing results against blank controls and negative controls (*Orientia tsutsugamushi*, Seoul *Orthohantavirus*), along with positive controls.

### 2.9. qPCR Repeatability and Stability Evaluation

Six concentration gradients (1.00 × 10^4^ ~1.00 × 10^9^ copies/µL) of positive standards were utilized as templates. Intra-group repetitions were conducted, with each concentration repeated thrice. This operation was performed once a week, totaling three inter-group repetitions. RNase-free ddH_2_O served as the negative control group. The mean C_t_ values, standard deviation (SD), and coefficient of variation (CV) were computed to evaluate reproducibility and stability.

### 2.10. Comparison of qPCR and Conventional PCR Assays

Conventional PCR reactions were executed using serial ten-fold gradient dilutions of positive standards as positive templates, with RNase-free H_2_O as the negative control. Amplified PCR products underwent agarose gel electrophoresis to compare the sensitivity of qPCR with conventional PCR methods.

### 2.11. qPCR for Small Mammal Samples

Small mammal samples underwent DNA extraction from tissues using the aforementioned method. Subsequently, all samples were subjected to *Bartonella* assay following optimized qPCR experimental conditions.

### 2.12. Tissue Tropism of Bartonella spp. in Small Mammals

The qPCR method was employed to detect positive samples and quantitatively analyze various tissues from small mammalian samples that tested positive for *Bartonella*. This approach aimed to investigate the tissue tropism of the pathogen.

### 2.13. Statistical Test

Analysis of the data was carried out utilizing one-way analysis of variance (ANOVA) with GraphPad Prism software version 8.0 (GraphPad Software, San Diego, CA, USA). All results are expressed as the mean ± standard error of the mean (SEM). We determined statistical significance using *p*-values: *p*-values below 0.05 were considered statistically significant, while those below 0.01 were deemed highly significant.

## 3. Results

### 3.1. Sample Collection and Testing

Notably, Chevrier’s field mouse (*Apodemus chevrieri*) and the small Oriental vole (*Eothenomys eleusis*) emerged as the dominant species, constituting 27.4% (*n* = 94) and 44.0% (*n* = 151), respectively. The samples (*n* = 333) underwent *Bartonella* spp. testing via qPCR, revealing a natural infection rate of 31.5% (*n* = 105, 95% CI: 27.7–36.7%). The prevalencein Chevrier’s field mouse (*Apodemus chevrieri*), the small Oriental vole (*Eothenomys eleusis*), the Chestnut white-bellied rat (*Niviventer fulvescens*), the Asian house rat (*Rattus tanezumi*), the long-tailed red-toothed shrew (*Episoriculus leucops*), the Chinese mole shrew (*Anourosorex squamipes*), and the Tibet pika (*Ochtona Thibetana*) were 44.4% (67/151), 27.7% (26/94), 100.0% (1/1), 6.3% (2/32), 60.0% (3/5), 60.0% (3/5), and 22.2% (2/9), respectively (Table 1). Additionally, the prevalence of *ssrA*, *rpoB*, and *gltA* in small mammals was 9.0% (*n* = 30), 2.9% (*n* = 10), and 1.4% (*n* = 5), respectively, as determined by conventional PCR. The qPCR demonstrated significantly higher detection rates than conventional PCR (χ^2^ = 34.8, *p* < 0.05), with *ssrA* showing a higher detection rate than *rpoB* and *gltA* (*p* < 0.05).

### 3.2. Comparison of Similarities among ssrA, rpoB, and gltA Genes in Bartonella spp.

In this study, 30 strains of *Bartonella* spp. were identified based on the *ssrA* gene. When compared with the nucleotide sequences of *Bartonella* sp. in GenBank, the homology ranged from 96.0% to 99.5%. Notably, nine strains from *A. chevreri* and four strains from *E. eleusis* exhibited a similarity of 96.0% to 97.0% with *B. tribocorum* in Guangdong Province, China. Furthermore, nine strains from *A. chevrieri* demonstrated a similarity ranging from 96.3% to 99.3% to *Bartonella* spp. Coyote22sub2 from blood samples of coyotes (*Canis latrans*) in California, USA. Additionally, four strains from *E. eleusis* showed up to 99.0% to 99.5% similarity to *Bartonella* spp. AR 15-3 in Switzerland. Moreover, three strains from *Anourosorex. squamipes*, *R. tanezumi*, and *A. chevrieri* were found to be 96.6%, 97.7%, and 98.0% similar, respectively, to *B. taylorii* from blood samples of *Microtus* spp. in France. Lastly, one strain from *N. fulvescens* exhibited 97.7% similarity to uncultured *Bartonella* spp. from the Natal multimammate mouse (*Mastomys natalensis*) in Tanzania (Appendix A).

For the *rpoB* gene of *Bartonella* spp., 10 strains were obtained, and when compared with *Bartonella* spp. sequences in GenBank, the nucleotide levels were homologous to a range of 95.8% to 100.0%. Notably, two strains from *A. chevrieri* and two strains from *E. eleusis* showed a similarity of 86.7% to 100.0% to *B. grahamii* from *A. chevrieri* in Yunnan Province, China. Furthermore, one strain from *A. chevrieri* exhibited the highest similarity of 96.4% to *B. rochalimae* from the blood of a dog in Japan. Another strain from *A. chevrieri* and one from *E. eleusis* demonstrated the highest similarity of 95.8% to 96.9% with B. Sendai from *Microtus montebelli* in Japan. Additionally, two strains from *A. chevrieri* and one from *E. eleusis* were 97% similar to *B. koshimizu* from the striped field mouse (*A. agrarius*) in Japan, while one strain from *N. fulvescens* showed 97.6% similarity to *B. phocinensis* from *R. norvegicus* in France (Appendix A).

Concerning the *gltA* gene of *Bartonella* spp., five strains were obtained, and the nucleotide similarity with *Bartonella* spp. sequences in GenBank ranged from 92.1% to 99.9%. Notably, two strains from *A. chevrieri* had the highest similarity of 99.7% to 100.0% with *B. koshimizu* from *A. agrarius* in Japan. Additionally, one strain from *A. chevrieri* exhibited the highest similarity of 97.4% with *B. grahamii* from the social vole (*Microtus socialis*) in Georgia. Furthermore, one strain from *E. leucops* showed 96.3% similarity to uncultured *Bartonella* sp. from the black-legged tick (*Ixodes. scapularis*) in the USA. Interestingly, one strain (GS136) from the long-tailed shrew (*E. leucops*) had the highest similarity of only 92.1% with uncultured *Bartonella* sp. from fleas (*Ctenophthalmus lushuiensis*) in China, potentially representing a new species of *Bartonella* (Appendix A, Appendix A).

### 3.3. Phylogenetic Analysis

In the phylogenetic tree constructed based on the *ssrA* gene, the 30 sequences are classified *Bartonella*, forming four distinct categories. The first branch comprises HQ92, HQ38, HQ76, and the six folded sequences (HQ66, HQ72, HQ73, HQ77, HQ96, and HQ99) in unclassified *Bartonella* spp. The second branch comprises GS183, GS167, and GS55 in *B. taylorii*. The third branch comprises GS164, HQ9, and twelve folded sequences (HQ4, HQ18, HQ25, HQ32, HQ34, HQ35, HQ55, HQ80, GS29, GS35, GS37, GS60) in *B. tribocorum*. The fourth branch consists of GS39, GS40, GS43, and GS97 in *B. rochalimae* (Figure 2).

In the phylogenetic tree based on the *rpoB* gene, ten sequences form five distinct clades. HQ45 and HQ65 from *A. chevrieri* and GS8 from *E. eleusis* cluster into *B. grahamii*. HQ38 from *A. chevrieri* forms a clade with *B. rochalimae*. GS109 from *E. eleusis* and GS2 from *A. chevrieri* group together in *B. Sedai*. GS9, HQ9, and HQ25 from *A. chevrieri* cluster into *B. koshimizu*. Finally, GS164 from *N. fulvescens* groups into *B. phoceensis* (Figure 3).

In the phylogenetic tree based on the *gltA* gene, five sequences are categorized into three groups. The first branch includes GS134 from *E. leucops* and HQ64 from *A. chevrieri*, clustering with *B. grahamii*. The second branch comprises GS136 from *E. leucops*, aligning with *B. taylorii*. The third branch consists of GS129 and HQ19 from *A. chevrieri*, forming a clade with *B. koshimizu* (Figure 4).

### 3.4. Establishment of qPCR Standard Curve

To construct the qPCR standard curve, five pairs of *Bartonella* standard products ranging from 1.00 × 10^4^ to 1.00 × 10^8^ copies/µL were selected for the X-axis, with the corresponding C_t_ values plotted on the Y-axis. The resulting graph demonstrates a robust linear relationship between the dilution templates and C_t_ values (Appendix A).

### 3.5. Evaluation of qPCR Sensitivity and Specificity

The study revealed that the minimum detectable copy number of *Bartonella* spp. positive standard was 1.00 × 10^2^ copies/µL, indicating the excellent sensitivity of the established qPCR method (Appendix A). In this investigation, testing was conducted with *Bartonella*, *Orientia tsugamushi*, and Seoul *Orthohantavirus*. The primer employed could exclusively amplify *Bartonella*.

### 3.6. Evaluation of qPCR Repeatability and Stability

Inter-group repeatability experiments demonstrated that the standard deviation of *Bartonella* standard detection within each concentration group ranged from 0.06 to 0.40, and the coefficient of variation varied between 0.53 and 1.84. These results affirm the robust reproducibility of the established qPCR method (Appendix A). Intra-group repeatability experiments further revealed that the standard deviation of *Bartonella* standard detection at various concentrations between groups ranged from 0.19 to 0.55, with a coefficient of variation between 0.82 and 1.92. These findings attest to the stability of the established qPCR method (Appendix A).

### 3.7. Comparison of Sensitivity between qPCR and Conventional PCR Methods

The experimental outcomes indicated that the lowest copy number for positive standard detection of *Bartonella* by qPCR was 1.00 × 10^2^ copies/µL. In contrast, the lowest copy number for *Bartonella* positive standards detected by conventional PCR was 1.00 × 10^3^ copies/µL (Appendix A).

### 3.8. Tissue Tropism of Bartonella spp. in Small Mammals

A total of 30 *Bartonella*-positive samples were identified using the *ssrA* gene. The quantitative analysis of naturally infected *Bartonella* in various tissues, including the heart, liver, spleen, lung, kidney, intestine, and brain, was conducted using the qPCR method. The mean copy number of *Bartonella* was most pronounced in spleen tissue across all positive samples, registering at 3.37 × 10^5^ copies/g, indicating a notably higher copy number of *Bartonella* in the spleen. The mean copy numbers in other tissues were as follows: spleen (3.36 × 10^5^), heart (1.17 × 10^5^), brain (9.70 × 10^4^), kidney (2.56 × 10^4^), lung (2.50 × 10^4^), liver (2.29 × 10^4^), and intestine (2.00 × 10^4^) copies/g, respectively (Table 2).

Notably, spleen tissues exhibited significantly higher copy numbers of *Bartonella* compared to kidney and intestinal tissues, and differences in *Bartonella* copy numbers were observed across all tissues (*p* < 0.0001). Specific variations in *Bartonella* copy numbers were identified in certain tissues (Figure 5).

## 4. Discussion

The current study detected *Bartonella* in seven species: *A. chevrieri*, *E. eleusis*, *N. fulvescens*, *R. tanezumi*, *E. leucops*, *A. squamipes*, and *O. thibetana*. *Bartonella* can infect a broad range of hosts. Among these, *B. grahamii*, *B. rochalimae*, and *B. koshimizu* were detected in *A. chevrieri*; *B. grahamii*, *B. sendai*, and *B. koshimizu* were detected in *E. eleusis*; and *B. phoceensis* was detected in *N. fulvescens*. The uncultured *Bartonella* spp. was detected in *R. tanezumi* and *A. squamipes*.

The study investigated the natural infection status and molecular characteristics of *Bartonella* species carried by small mammals in two counties of Yunnan Province. This is the first reported survey of *Bartonella* species in these areas. According to the results, the prevalence of *Bartonella* is lower compared to certain regions in China, such as Yunnan Province (57.7%) [33], Qinghai Province (38.61%) [34], Heilongjiang Province (57.7%) [35], and Shanxi Province (49.50%) [36]. However, it is higher than in Zhejiang Province (31.4%) [37] and Fujian Province (16.19%) [38]. Regions at higher latitudes tend to have a higher infection rate of *Bartonella*. The latitude of the area studied is relatively low, but there is a high density of rodents and a high species richness [39]. This accounts for the relatively high infection rate in our study. In addition, compared to conventional PCR results, the infection rate obtained through qPCR detection is higher. qPCR can detect samples with low copy numbers, providing a more accurate reflection of the infection situation. Using qPCR is recommended for investigating *Bartonella*.

The *ssrA* gene, along with the *gltA* and *rpoB* genes, possesses identical species identification functions [29]. The *ssrA* can also be employed for the swift detection and classification of *Bartonella*. According to the DNA sequences determined by *gltA*, *rpoB*, and *ssrA* genes, at least seven species of *Bartonella* were detected in these small mammals through homology and phylogenetic analysis. These include *B. grahamii*, *B. rochalimae*, *B. sendai*, *B. koshimizu*, *B. phoceensis*, *B. taylorii*, *B. koshimizu*, and a newly isolated *Bartonella* (GS136: the highest similarity of only 92.1% with uncultured *Bartonella* spp.) that should be considered a new species based on the *gltA* fragment of 327 bp having 96.0% sequence similarity to the validated species or the *rpoB* fragment of 825 bp having 95.4% similarity to the sequence of the evidenced species [40]. In addition, *B. grahamii*, originally isolated from the bank vole (*Myodes glareolus*) in the UK and subsequently linked to retinitis and cat-scratch disease (CDS) [2,41,42,43], can infect humans. These results indicate the presence of diverse *Bartonella* species among small mammals in the survey areas that can cause human disease.

The qPCR method demonstrates excellent sensitivity, specificity, stability, and repeatability. Quantitative studies have revealed that the copy number of *Bartonella* species in spleen tissues is significantly higher in infected small mammals (*p* < 0.0001), indicating the splenic tissue tropism of infected small mammals. *Bartonella* is extremely difficult to culture in vitro due to harsh nutritional conditions, but since *Bartonella* attacks endothelial cells to cause bacteremia, blood is used for in vitro culturing. During sampling, blood is more difficult to collect, and tissue isolation is relatively simple, so the spleen tissue can be the organ of choice for tissue isolation. Variable DNA copy numbers were also detected in other tissues, suggesting that *Bartonella* species have broad tissue tropism.

Unfortunately, we lack blood samples, and only partial brain tissue samples are available. Nevertheless, we detected three unclassified *Bartonella* in ten positive brain tissues, responding quantitatively to the presence of a large number of *Bartonella* in brain tissues. *B. henselae* and *B. quintana* have been reported to cause severe central nervous system disease [44,45]. The antibodies and DNA of *Bartonella* have been found in the cerebrospinal fluid of cats and dogs, and *Bartonella* has also been isolated from rodent brains [34]. However, understanding how *Bartonella* enters brain tissue and the mechanisms through which it affects the central nervous system requires further study.

## 5. Conclusions

In conclusion, this study identified seven *Bartonella* species, namely, *B. grahamii*, *B. rochalimae*, *B. sendai*, *B. koshimizu*, *B. phoceensis*, *B. taylorii*, and a newly isolated *Bartonella* (GS136), within seven small mammal species: Chevrier’s field mouse (*A. chevrieri*), the small Oriental vole (*E. eleusis*), the Chestnut white-bellied rat (*N. fulvescens*), the Asian house rat (*R. tanezumi*), the long-tailed red-toothed shrew (*E. leucops*), the Chinese mole shrew (*A. squamipes*), and the Tibet pika (*O. thibetana*). Our study also highlighted the superior detection efficiency of qPCR compared to conventional PCR, with the qPCR method demonstrating higher sensitivity. Furthermore, the detection rate of *Bartonella* and the *ssrA* gene surpassed that of the *rpoB* gene and *gltA* gene, emphasizing the efficacy of utilizing the *ssrA* gene for rapid and accurate *Bartonella* detection and classification. *Bartonella* exhibited a natural ability to infect various tissues in small mammals, with brain tissue being among the affected organs. The broad tissue tropism observed, particularly with the highest load in spleen tissue, suggests the systemic nature of *Bartonella* prevalence. These findings contribute to our understanding of the diversity, prevalence, and tissue tropism of *Bartonella*, shedding light on the potential public health implications associated with these infections.

## Figures and Tables

**Figure 1 animals-14-01320-f001:**
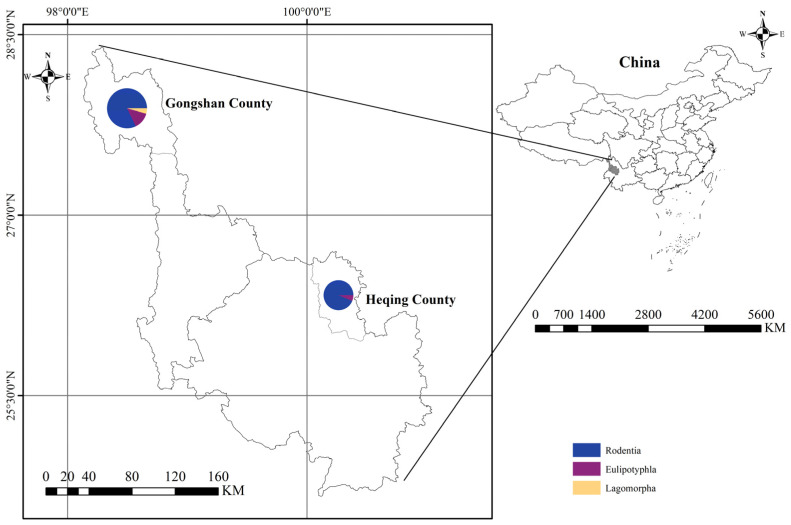
Sampling locations. The right figure depicts a map of China, while the left figure illustrates maps of Heqing County and Gongshan County, located in Yunnan Province, China.

**Figure 2 animals-14-01320-f002:**
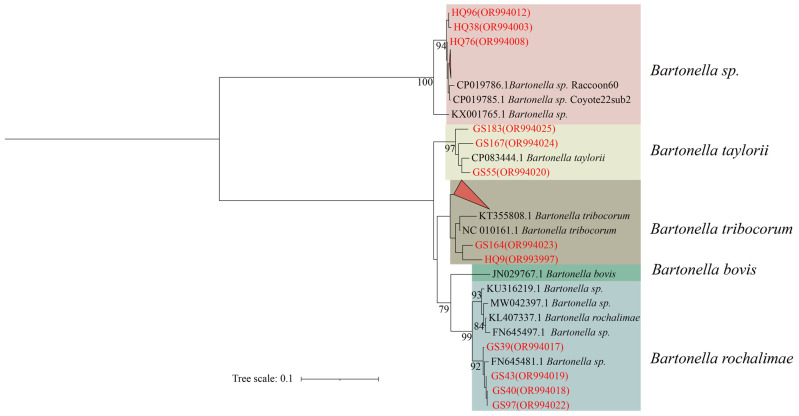
Phylogenetic tree based on the *ssrA* gene in small mammals from Yunnan Province, China. Note: The red triangle above represents: HQ66, HQ72, HQ73, HQ77, HQ92, HQ99; the red triangle below represents: HQ4, HQ18.

**Figure 3 animals-14-01320-f003:**
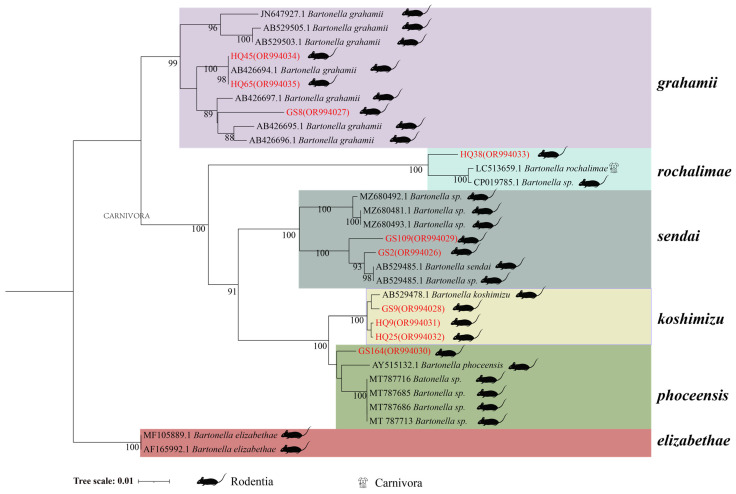
Phylogenetic tree based on the *rpoB* gene in small mammals from Yunnan Province, China. HQ25, HQ32, HQ34, HQ35, HQ55, HQ80, GS29, GS35, GS37, GS60.

**Figure 4 animals-14-01320-f004:**
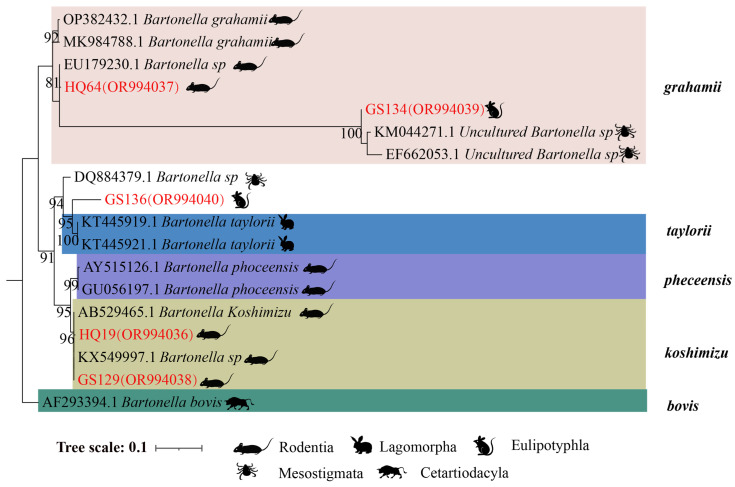
Phylogenetic tree based on the *gltA* gene in small mammals from Yunnan Province, China.

**Figure 5 animals-14-01320-f005:**
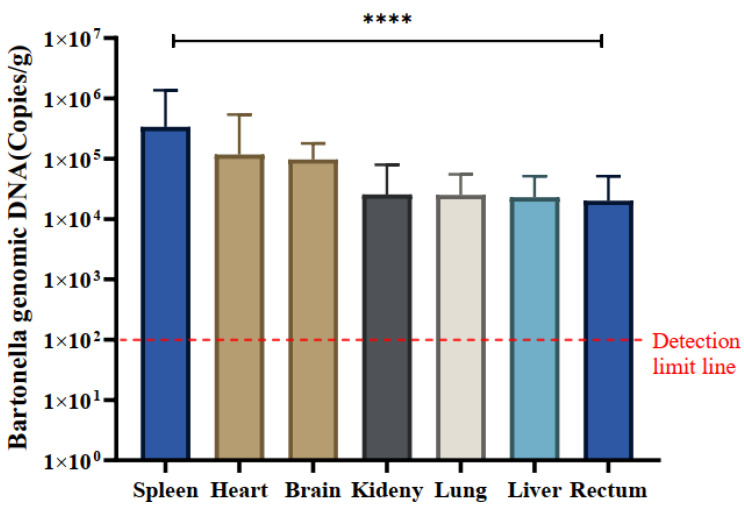
Tissue distribution of *Bartonella* species in positive samples. The quantification (mean ± standard error) of *Bartonella* species in the spleen, heart, brain, kidney, lung, liver, and rectum of 30 positive samples was measured in copies/g. The significance test was performed using one-way ANOVA (**** *p* < 0.0001). The detection limit of qPCR is indicated by a red dashed line in this study.

**Table 1 animals-14-01320-t001:** Molecular prevalence of *Bartonella species* in small mammals from Heqing County and Gongshan County, Yunnan Province.

Order	Species	Locations (County)	Composition (%)	Prevalence, %			
				qPCR	Conventional PCR		
					*ssrA*	*rpoB*	*gltA*
*Rodentia*	Chevrier’s field mouse (*Apodemus chevrieri*)	Heqing, Gongshan	151/333 (45.4)	67/151 (44.4)	23/151 (19.0)	5/151 (3.3)	3/151 (1.9)
	Small Oriental vole (*Eothenomys eleusis*)	Heqing, Gongshan	94/333 (28.2)	26/94 (27.7)	4/94 (4.2)	4/94 (4.2)	0/94 (0.0)
	Chestnut white-bellied rat (*Niviventer fulvescens*)	Heqing, Gongshan	1/333 (0.3)	1/1 (100.0)	1/1 (100.0)	1/1 (100.0)	0/1 (0.0)
	Asian house rat (*Rattus tanezumi*)	Heqing, Gongshan	32/333 (9.6)	2/32 (6.3)	3.1 (1/32)	0/32 (0.0)	0/32 (0.0)
	Norway rat (*Rattus norvegicus*)	Gongshan	1/333 (0.3)	0/1 (0.0)	0/1 (0.0)	0/1 (0.0)	0/1 (0.0)
	White-footed Indochinese rat (*Rattus nitidus*)	Heqing	1/333 (0.3)	0/1 (0.0)	0/1 (0.0)	0/1 (0.0)	0/1 (0.0)
	Ryukyu mouse (*Mus caroli*)	Heqing	3/333 (0.9)	0/3 (0.0)	0.0 (0/3)	0/3 (0.0)	0/3 (0.0)
	House moue (*Mus musculu*)	Heqing	2/333 (0.6)	0/2 (0.0)	0.0 (0/2)	0/2 (0.0)	0/2 (0.0)
	Swinhoe’s striped squirrel (*Tamiops swinhoei*)	Heqing	1/333 (0.3)	0/1 (0.0)	0/1 (0.0)	0/1 (0.0)	0/1 (0.0)
	Smoke-bellied niviventer (*Niviventer eha*)	Gongshan	2/333 (0.6)	0/2 (0.0)	0/2 (0.0)	0/2 (0.0)	0/2 (0.0)
*Eulipotyphla*	Long-tailed red-toothed shrew (*Episoriculus leucops*)	Gongshan	5/333 (1.5)	3/5 (60.0)	0/5 (0.0)	0/5 (0.0)	2/5 (40.0)
	Asian gray shrew (*Crocidura attenuate*)	Gongshan	14/333 (4.2)	0/14 (0.0)	0/14 (0.0)	0/14 (0.0)	0/14 (0.0)
	Chinese mole shrew (*Anourosorex squamipes*)	Gongshan	17/333 (4.2)	4/17 (23.5)	1/17 (16.7)	0/17 (0.0)	0/17 (0.0)
*Lagomorpha*	Tibet pika (*Ochtona Thibetana*)	Gongshan	9/333 (2.7)	2/9 (22.2)	0/9 (0.0)	0/9 (0.0)	0/9 (0.0)
	Total		333/333				

**Table 2 animals-14-01320-t002:** *Bartonella* species copies in tissues of small mammals.

	Spleen	Heart	Brain	Kidney	Lung	Liver	Rectum
Mean (Copies/g)	3.3 × 10^5^	1.1 × 10^5^	9.7 × 10^4^	2.5 × 10^4^	2.5 × 10^4^	2.2 × 10^4^	2.0 × 10^4^
SEM (Copies/g)	1.8 × 10^5^	7.6 × 10^4^	1.5 × 10^4^	9.7 × 10^3^	5.4 × 10^3^	5.2 × 10^3^	5.7 × 10^3^

SEM: Standard error of the mean.

## Data Availability

The datasets used in this study can be accessed by contacting the corresponding author in a reasonable manner. All sequences referenced in this article are retrievable from the NCBI database (https://www.ncbi.nlm.nih.gov, accessed on 20 August 2023).

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
