# Peer review of "Molecular Prevalence, Genetic Diversity, and Tissue Tropism of Bartonella Species in Small Mammals from Yunnan Province, China"

_animals, 2024, doi:10.3390/ani14091320_

Round 1

Reviewer 1 Report

Comments and Suggestions for Authors

A strength of this manuscript is the comprehensive description of molecular techniques and methodology used in these specific studies.

However, the analysis of one animal in a group (Niviventer fulvescens)  and reporting that this represents 100% infection for that species is misleading.  

The major flaw is the lack of a description about how these results could impact the public health for people living in this region of China.  

Line 17 wording not clear

Line 31 wording not clear

Line 33 wording not clear

Line 37 widely parasitic is an odd wording not sure what is meant by this

Lines 77-78 references needed

Last line of Table 1 (30/33) should be 30/333?

Figure 2,3, 4  very difficult to read due to colors, red is difficult to read in all Phylogenetic tree figures, abbreviations are not clearly defined

Line 474 These findings (need to reword this sentence or leave it out)

Comments on the Quality of English Language

English is very good, some small suggestions above.

Reviewer 2 Report

Comments and Suggestions for Authors

The Authors studied the diffusion and the species of Bartonella among small mammals in the chinese province of Yunnan, using conventional and quantitative PCR, last one for detecting tissue bacterial distributions in different organs. The work is well written, the areas of major strenght are the extensive descriptions of matherials, methods and results while the discussion could be expanded precisely in relation to the amount of data obtained (for example analysing in more depth the pathogenic value of each species of Bartonella for man and animals or a possible explanation of the Bartonella prevalence in spleen tissue compared to other organs).

I suggest however a minor revision for the following points:

Line 19 and 260 (and everywhere you use this definition): for avoiding confusion about the health status of the small mammals collected (if sick or only carrier), it could be better to refer to “Bartonella prevalence” and not “infection”.

Line 25, Table 1, lines 307, 315, 319,321, 326,344, 419, 439, 470, 472 (check also elsewhere in the text): the species all in small letters (for example Eothenomys eleusis).

Line 37: I suggest to start with “Bartonella genus constitues a group…” eliminating genus Bartonella at the end in line 39-40.

Line 95: as the animals collected are the samples/materials studied (and not a result of the study), I suggest to move lines 255-257 here, in Material and Methods chapter, inserting the reference atlas book (or other) that supported the identification of different species of small mammals.

Line 102: describe the method (pharmacological?) used for sacrificing the animals.

Table 1: change Composition in “N° (%)” modifying in the Table as for example “151/333 (45.4%)”; the number and not the prevalence of each species among the group of animals tested is relevant for the topic of the paper (also in order to evaluate, for example, the statistical significance of 100% prevalence qPCR in Chestnut white-bellied rat, calculated with only one animal tested).

Table 3: insert in the legend the explanation of the various animal symbols used after the Bartenella species.

Line 457: it’s not clear the relation between “altough we conducted…” and the “majority of samples are well preserved”: how does the quantitative analysis compromise the quality of tissues?
